# Giant viruses specific to deep oceans show persistent presence and activity

Wenwen Liu,[1] Komei Nagasaka,[1] Junyi Wu,[1] Hiroki Ban,[2] Ethan Mimick,[3] Lingjie Meng,[1] Russell Y. Neches,[1] Mohammad Moniruzzaman,[3] Takashi Yoshida,[4] Yosuke Nishimura,[5] Hisashi Endo,[1] Yusuke Okazaki,[1] Hiroyuki Ogata[1]

**ABSTRACT** Giant viruses (GVs) of the phyla *Nucleocytoviricota* and *Mirusviricota* are large double-stranded DNA viruses that infect diverse eukaryotic hosts and impact biogeochemical cycles. Their diversity and ecological roles have been well studied in the photic layer of the ocean, but less is known about their activity, population dynamics, and adaptive strategies in the aphotic layers. Here, we conducted eight seasonal time-series samplings of the surface and mesopelagic layers at a coastal site in Muroto, Japan, and integrated 18S metabarcoding, metagenomic, and metatranscriptomic data to investigate mesopelagic GVs and their potential hosts. The analysis identified 48 GV genomes including six that were exclusively detected in the mesopelagic layer. Notably, these mesopelagic-specific GVs showed persistent activity across seasons. To further investigate the distribution and phylogenomic features of GVs at a global scale across broader depths, we compiled 4,473 species-level GV genomes from the OceanDNA MAG project and other resources and analyzed 1,890 marine metagenomes. This revealed 101 deep-sea-specific GVs, distributed across the GV phylogenetic tree, indicating that adaptation to deep-sea environments has occurred in multiple lineages. One clade enriched with deep-sea-specific GVs included a GV genome identified in our Muroto data, which displayed a wide geographic distribution. Seventy-six KEGG orthologs and 74 Pfam domains were specifically enriched in deep-sea-specific GVs, encompassing functions related to the ubiquitin system, energy metabolism, and nitrogen acquisition. These findings support the scenario that distinct GV lineages have adapted to hosts in aphotic marine environments by altering their gene repertoire to thrive in this unique habitat.

**IMPORTANCE** Giant viruses are widespread in the ocean surface and are key in shaping marine ecosystems by infecting phytoplankton and other protists. However, little is known about their activity and adaptive strategies in deep-sea environments. In this study, we performed metagenomic and metatranscriptomic analyses of seawater samples collected from a coastal site in Japan and discovered giant virus genomes showing persistent transcriptional activity across seasons in the mesopelagic water. Using a global marine data set, we further uncovered geographically widespread and vertically extensive groups of deep-sea-specific giant viruses and characterized their distinctive gene repertoire, which likely facilitates adaptation to the limited availability of light and organic compounds in the aphotic zone. These findings expand our understanding of giant virus ecology in the dark ocean.

**KEYWORDS** giant viruses, deep ocean, the Kochi Prefectural Deep Seawater Laboratory, metatranscriptome, adaptation

G iant viruses (GVs), encompassing the phylum *Nucleocytoviricota* and the recently proposed phylum *Mirusviricota*, are characterized by their large virion and genome

**Peer Reviewer** Hanpeng Liao, Fujian Agriculture and Forestry University, Fuzhou, China

Address correspondence to Hiroyuki Ogata, ogata@kuicr.kyoto-u.ac.jp.

The authors declare no conflict of interest.

See the funding table on p. 16.

sizes (1–3). They infect a broad range of eukaryotes from unicellular algae, diverse heterotrophic protists, to larger animals (4–6). In marine environments, GVs affect the population dynamics of their hosts including bloom-forming algae (7–9), alter host metabolism via virus-induced metabolic reprogramming (10, 11), and consequently impact carbon and nutrient cycles (12). Previous metagenomic studies have revealed the high abundance (13), activity (14, 15), diversity (16–18), and widespread geographic GV distribution in the ocean (19–21). The community structure of GVs in marine environments tightly correlates with that of microeukaryotes in time (22–24) and space (25), supporting the idea that GVs regulate the microeukaryote community.

To date, studies have predominantly focused on the ocean's photic layer, while aphotic layers—which constitute approximately 95% of the global ocean (26)—remain comparatively underexplored owing to the inherent difficulty in accessing these environments. Of note, all marine GVs have been isolated from the photic layer, except for one case from deep-sea sediment (27). Several previous studies, nevertheless, detected the presence of GV-related sequences in deep-sea environments, including genomic contigs (28–30) and conserved marker genes (19, 31). Another study reported the transcription signals from mirusvirus genomes in mesopelagic waters based on *Tara* Oceans data (2). More recently, five GV metagenome-assembled genomes (GVMAGs) were shown to be unique to the deep water below 150 m (32), and a few GVMAGs were specifically and highly abundant in the deep layers of the North Pacific Subtropical Gyre (30). These results clearly indicate the existence of GVs in deep oceans. However, whether these GVs represent deep-sea residents actively infecting sympatric hosts or are just passively transported from the surface community through sinking remains unclear. For example, Endo et al. showed that the GV lineages detected in the mesopelagic layer are mostly (99%) also present in the photic layer and suggested that they are largely settled from the photic layer (19). Sheam et al. also provided evidence that some GVs detected in sediment trap samples (4,000 m depth) were vertically transported from the surface layer (30). Nonetheless, a recent study revealed the existence of a GV community specific to an aphotic layer (65 m depth) of a freshwater lake (33), implying the existence of marine GVs specifically inhabiting dark environments.

Investigating GVs in aphotic layers is challenging because of methodological and logistical difficulties (34), as GVs have lower abundance than bacteria in the same size fraction, especially in deeper layers (13). Thus, a large volume of water must be sampled to concentrate the GVs. Furthermore, collecting samples for transcriptomic analysis to probe viral activity needs to be conducted in a relatively short time to avoid RNA degradation and shifts in the physiological state of the microorganisms (35). The Kochi Prefectural Deep Seawater Laboratory, located in Muroto, Kochi Prefecture, Japan, is situated at a coastal site, where oligotrophic water influenced by the Kuroshio Current can be readily accessed (36). The laboratory has the facility to continuously draw untreated surface (0.5 m) and mesopelagic (320 m) seawater (37) and supports scientific research of deep-sea water and various industrial applications such as food processing and aquaculture (38).

In this study, we applied metabarcoding, metagenomic, and metatranscriptomic analyses of the surface and mesopelagic samples collected across seasons at the Laboratory in Muroto to investigate the existence and activity of GVs, as well as the sympatric potential host community, in the mesopelagic layer. We identified and characterized 48 GVMAGs, 6 showing stable activity exclusively in the mesopelagic water. We then compiled a global GVMAG data set, named the Giant Virus Genome Reference (GVGR) database, from publicly available data (2, 6, 23, 33, 39, 40), and performed a systematic analysis to identify deep-sea-specific GVs at a global scale. The analysis revealed 101 GVMAGs preferentially detected in deep oceans, some forming deep-sea-specific GV clades in the phylogenetic tree.

## RESULTS AND DISCUSSION

### Data overview

The pump systems at the study site (i.e., off the coast of Muroto; Fig. S1) enabled us to collect and filter a large volume of water from two depths (0.5 m and 320 m) in a short time (Table S1). Size fractionations were performed to concentrate different microorganisms: pico-fraction (0.2–3.0 µm or 0.2–5.0 µm), total-fraction (0.2–150 µm), and nano/micro-fraction (3.0–150 µm or 5.0–150 µm) (Table S2). From these samples, we successfully extracted adequate quantities (>500 ng) of high-quality DNA and RNA for sequencing (Table S3). We generated 18S rDNA and rRNA metabarcodes (4,981 ASVs; nano/micro-fraction and either pico- or total-fraction) (Table S4), metagenomic data (757 Gbp; pico- or total-fraction), and metatrancriptomic data (598 Gbp; nano/micro-fraction and either pico- or total-fraction).

### Mesopelagic microeukaryote community is distinct from that in the surface water at the sampling site

In this study, 18S rDNA metabarcodes were used to measure the relative abundance of microeukaryotes at the ASV level although the abundance of 18S rDNA is biased by the copy number of rRNA genes, which varies among organisms (41, 42). 18S rRNA metabarcodes were used as a proxy for the metabolic activity of organisms (43). The microeukaryote community in the mesopelagic layer substantially differed from that of the surface layer at the study site (Fig. 1a through d), suggesting that the two layers harbor distinct host candidates for GVs. Furthermore, the mesopelagic microeukaryote community characterized by 18S rRNA differed from that of 18S rDNA metabarcodes, suggesting that some microeukaryotes are active in the mesopelagic layer, while others are inactive (i.e., in a dormant state or dead cells settled from the surface). For instance, Polycystinea had the highest abundance but was ranked 11th in activity (Fig. S2). Spirotrichea did not show high abundance but ranked third in activity (Fig. S2). These results suggest that the mesopelagic communities are a mixture of active and inactive organisms.

Non-metric multidimensional scaling (NMDS) ordination confirmed a clear separation between the surface and mesopelagic microeukaryote communities, regardless of whether 18S rDNA (Fig. 1e; $R^2$ = 0.25, PERMANOVA, $P$-value < 0.01) or 18S rRNA (Fig. 1f; $R^2$ = 0.24, PERMANOVA, $P$-value < 0.01) metabarcodes were used. Furthermore, the mesopelagic microeukaryote communities showed less enhanced seasonal changes than the surface communities. To compare the persistence of ASVs between the layers, we calculated Levins' standardized niche breadth (BA) using only the surface and mesopelagic samples collected on the same day. In both the 18S rDNA and 18S rRNA data sets, the BA values were significantly higher for the mesopelagic than the surface ASVs in both size fractions (Fig. S3, $P$-value < 0.001), indicating higher persistence for the mesopelagic ASVs. These results suggest that the mesopelagic environment may provide a unique and stable host community for GVs.

### GVs exclusively detected in mesopelagic metagenomes show persistency at the study site

Forty-eight GVMAGs recovered from the Muroto metagenomic data met our quality criteria (Fig. S4; see Materials and Methods), including 43 *Nucleocytoviricota* MAGs (NCV-MAGs) and 5 *Mirusviricota* MAGs (MV-MAGs) (Fig. 2a). These GVMAGs exhibited genome sizes ranging from 54 kbp to 824 kbp and GC contents ranging from 24% to 53% (Table S5). To assess the recovery level of GV lineages by GVMAGs, we investigated family-B DNA polymerase (PolB) sequences in the metagenomic data set. Through phylogenetic analysis (Fig. S5), we identified 163 PolB sequences predicted to be of GV origin. Of these PolB sequences, 34 (21%) were found in the Muroto GVMAGs, suggesting that many GVs in the study site were not represented in our GVMAG data set.

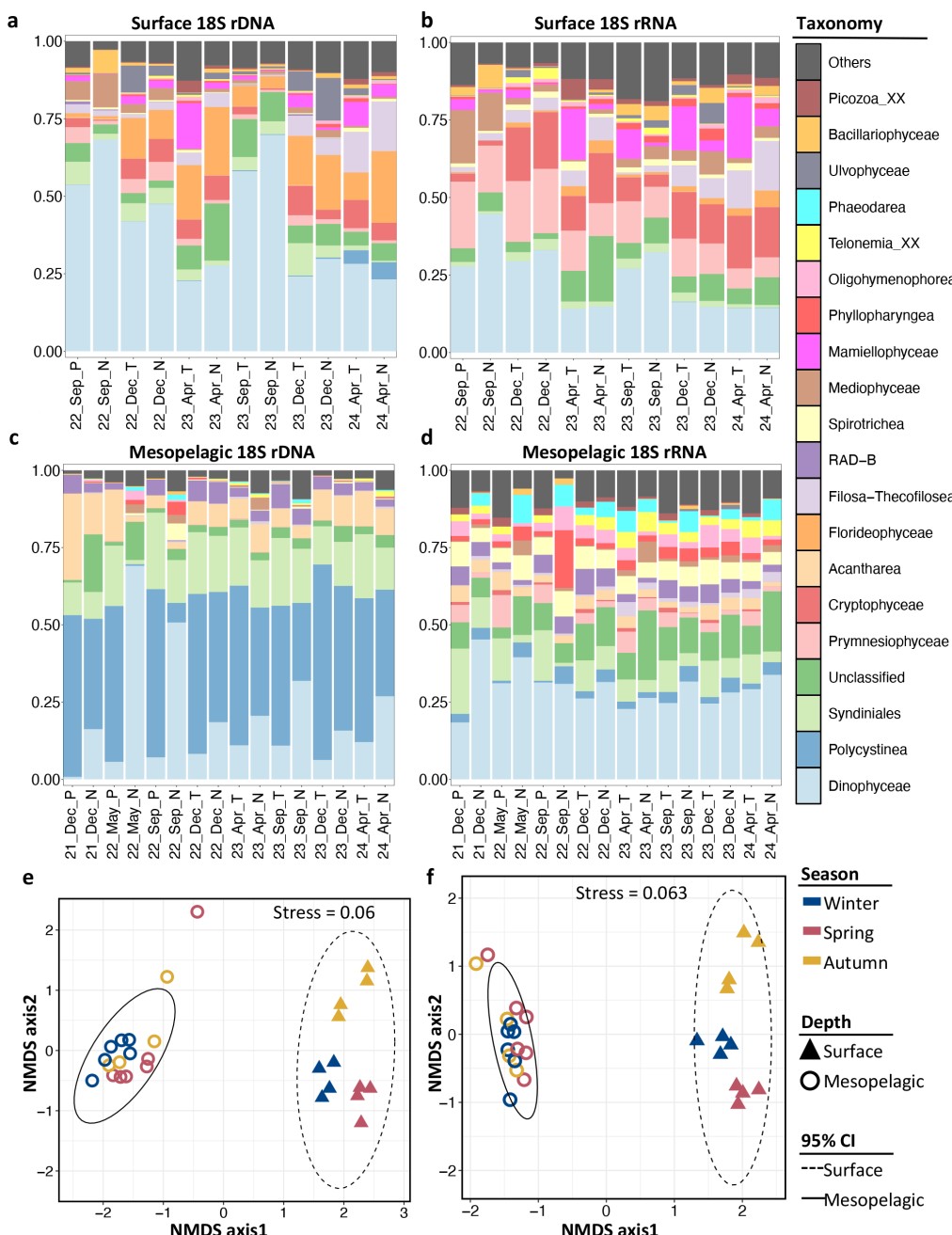

**FIG 1** Microeukaryote community composition of the seawater samples. (a–d) Relative abundance of taxa at the class level based on the metabarcoding data for surface 18S rDNA (a), surface 18S rRNA (b), mesopelagic 18S rDNA (c), and mesopelagic 18S rRNA (d). *X*-axis labels indicate the sampling date and size fractions (P: pico-fraction; T: total-fraction; N: nano/micro-fraction). (e and f) Non-metric multidimensional scaling (NMDS) plots showing differences in community compositions between surface and mesopelagic samples based on 18S rDNA (e) and 18S rRNA (f). CI denotes confidence interval.

Phylogenetic analysis of 7 conserved NCV marker genes revealed that the 43 NCV-MAGs belonged to 5 orders of *Nucleocytoviricota*: 19 imiterviruses, 15 algaviruses, 5 pimascoviruses, 2 pandoraviruses, and 2 asfuviruses (Fig. S6a). Two NCV-MAGs belonging to the order *Pandoravirales* were closely related to Emiliania huxleyi viruses (average nucleotide identity, ~85%). The five MV-MAGs were classified based on HK97 major capsid protein (MCP) phylogeny, including three in the MR2 marine clade and two in a clade composed of mirusvirus MAGs from a freshwater lake (Fig. S6b) (33).

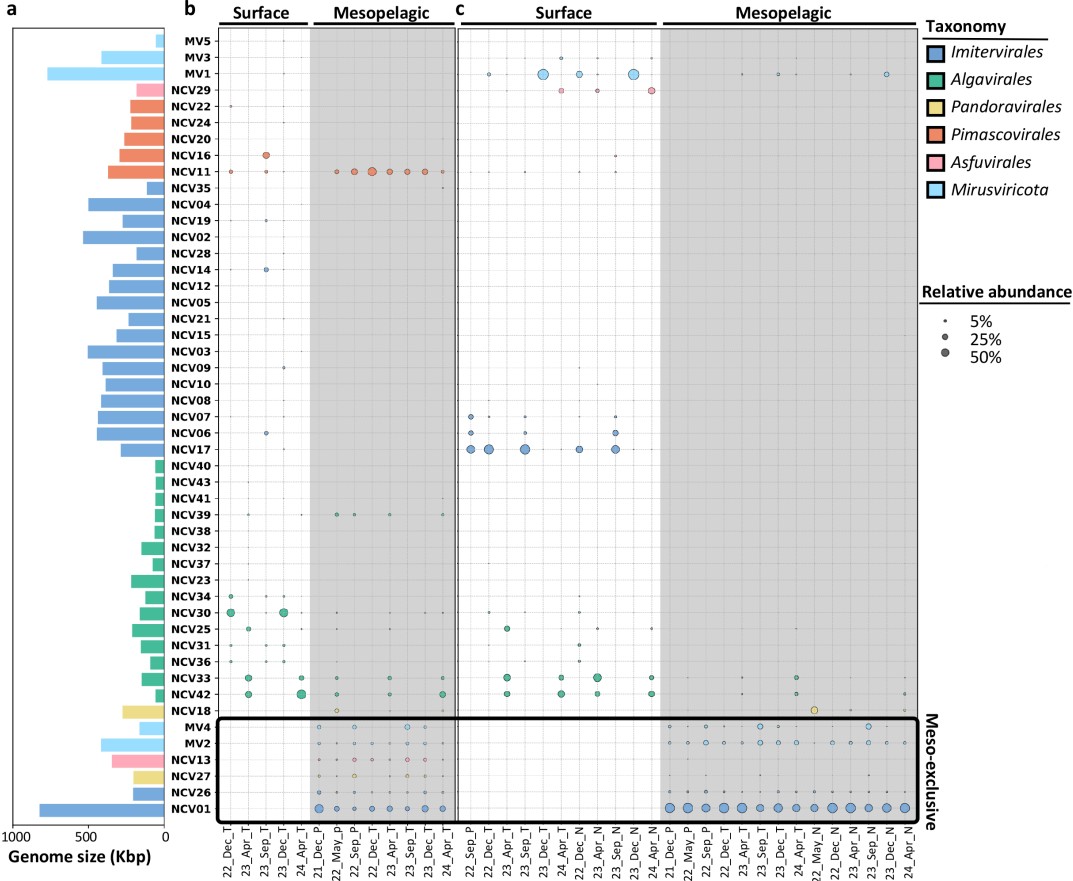

**FIG 2** Genome size distribution and relative abundance of Muroto GVMAGs in the metagenomic and metatranscriptomic data. (a) Genome size distribution of the 48 Muroto GVMAGs. (b) Relative abundance of Muroto GVMAGs in the metagenomic data. (c) Relative abundance of Muroto GVMAGs in the metatranscriptomic data. In (b) and (c), samples from the mesopelagic layer are shown with a gray background. Circle size (i.e., area) represents relative abundance, and circle color indicates taxonomic classification. GVMAGs enclosed by a black box are mesopelagic-exclusive GVMAGs. X-axis labels indicate the sampling date and size fractions (P: pico-fraction; T: total-fraction; N: nano/micro-fraction).

Mapping of the metagenomic reads on the 48 GVMAGs revealed that these MAGs represent 0.028%–1.28% (0.48% on average) of the total reads for the surface data compared with 0.01%–0.028% (0.019% on average) for the mesopelagic data (Fig. S7a). The lower relative abundance of GVs in the mesopelagic samples explains the lower level of GVMAG recovery from these samples. Among the 48 GVMAGs, 17 GVMAGs (15 NCV-MAGs, 2 MV-MAGs) were detected in the mesopelagic metagenomes, including six (NCV01, 13, 26, 27; MV2, 4) that were exclusively found in the mesopelagic samples (Fig. 2b; Table S6). These "meso-exclusive" GVMAGs tended to show persistent presence in the mesopelagic layer across different seasons (Fig. 2b). The other 11 GVMAGs detected in the mesopelagic metagenomic data were also detected in the surface metagenomes (Fig. 2b). In sharp contrast with the six meso-exclusive GVMAGs, many of these GVMAGs showed strong seasonal dynamics (e.g., NCV30, 42) akin to the pattern observed for GVs in a shallow coastal area (23). Endo et al. previously showed that the community structure of GVs in the mesopelagic layer (200–1,000 m) is different from that in the photic layer based on the *Tara* Oceans expedition data (19). Our results confirmed not only the distinct community structure but also the difference in the community dynamics between the surface and mesopelagic layers.

## Mesopelagic-specific GVs are active across seasons

Mapping of the metatranscriptomic reads on the GVMAGs revealed a similar trend to the metagenomic data (Fig. 2c; Fig. S7), with the meso-exclusive GVMAGs showing seasonally stable activity and the surface layer community showing strong seasonal dynamics. Seventeen GVMAGs showed transcriptional activity in the mesopelagic layer (Table S6). These included all six meso-exclusive GVMAGs (Fig. 2c). For five of these GVMAGs (NCV01, 26, 29; MV2, 4), transcripts were detected for over 10% of genes encoded in the genomes (Fig. S8), with NCV01, MV2, and MV4 expressing up to 37.3%–63.6% of their genes (Fig. S8). This result suggests ongoing viral replication in the mesopelagic environment. Furthermore, the transcriptional activities of many of these meso-exclusive GVMAGs were relatively constant over time (e.g., NCV01, NCV26, MV2, MV4) (Fig. 2c; Fig. S9). To the best of our knowledge, this is the first report of the transcriptional activity of GVs exclusively detected in an aphotic environment. The stable community structure and activity of meso-exclusive GVs across seasons parallel the stability of the microeukaryotic community in the mesopelagic zone at the study site (Fig. 1). In contrast to the meso-exclusive GVMAGs, several GVMAGs that were detected in both surface and mesopelagic metagenomes showed transcriptional activity only in the surface samples (e.g., NCV11, NCV30). These GVs in the mesopelagic layer may have originated from the surface through sinking processes, as previously suggested (19, 30). Overall, our findings support the existence of active and stable GV populations specific to the mesopelagic zone. Virus–host interactions may be less influenced by seasonal fluctuations in the relatively stable ecological conditions in this layer.

The difference in mapping depth for the metatranscriptome data between the surface and mesopelagic layers was less pronounced than the case for the metagenome data, with some sampling days showing comparable (or even higher) relative transcriptional activity of GVMAGs in the mesopelagic than the surface data. This suggests that even though fewer GVMAGs were reconstructed from the mesopelagic data, they may be highly active. For the total-fraction, the proportions of mapped transcriptomic reads were 0.0021%–0.54% (0.13% on average) for the surface samples and 0.0024%–0.0079% (0.0043% on average) for the mesopelagic samples. For the nano/micro-fraction, the proportions of mapped transcriptomic reads were 0.00028%–0.21% (0.052% on average) for the surface samples and 0.00024%–0.0017% (0.00074% on average) for the mesopelagic samples. The higher proportions of mapped mesopelagic transcriptomic reads in the total-fraction than in the nano/micro-fraction suggest that active infection of GVs in the mesopelagic layer is primarily associated with smaller host cells included in this fraction.

## Mesopelagic GVMAGs encode specific gene functions

The six meso-exclusive GVMAGs recovered in this study contained between 51 and 913 predicted genes per genome (Table S5), with 880 genes (35.5%) being assigned to 594 KO terms (Table S7). The six meso-exclusive GVMAGs showed distinct gene compositions enriched in functions related to virus–host interactions and signaling, including components of the ubiquitin system, cytoskeletal proteins, and signal transduction regulators.

Among genes in the ubiquitin system, E3 ubiquitin–protein ligase RNF181 was found in 23 GVMAGs. However, other ubiquitin system genes, such as Ariadne-1 (NCV01, NCV26, NCV27), E3 ubiquitin-protein ligase RNF13 (NCV01, NCV26), and NEDD4-binding protein 2 (NCV01, NCV26), were specific to meso-exclusive GVMAGs. Actin, myosin, and flagellar basal-body rod protein FlgG were exclusively present in NCV01 and NCV26 and could benefit viruses by manipulating the localization of viral replication machinery during infection (14, 44, 45). NCV01 also encoded ninein-like protein, which is a member of the γ-tubulin complexes binding proteins (46) and may function to modulate cytoskeleton. Additionally, phosphoinositide 3-kinase (PI3K), a regulator of membrane dynamics and endocytic trafficking, was detected in 4 of 6 meso-exclusive GVMAGs. PI3K

has been implicated in supporting the formation of replication organelles of enteroviruses and may similarly support GV replication (47).

Notably, among the top 20 transcriptionally active KO terms in the mesopelagic samples, 7 were related to the ubiquitin system (e.g., Ariadne-1, NEDD4-binding protein 2, ubiquitin B, and E3 ubiquitin-protein ligases RNF181, EDD1, SIAH1, RNFT1) (Fig. S10b), compared with only 1 in the surface samples (Fig. S10a). The sequential action of ubiquitin systems conjugates ubiquitin to proteins and target them to proteasomes for degradation (48). Eukaryotic viruses can hijack the ubiquitin-proteasome system to aid in various stages of viral propagation (49). Since normal protein folding is hindered by low temperature or high pressure (50, 51), the ubiquitin-related genes of GVs may function to modulate the host ubiquitin-proteosome system to quality control viral and host proteins in harsh condition in the mesopelagic layer, as previously hypothesized for GVs in cold environments (52). As the ubiquitin-proteasome system is also known to regulate apoptosis (53), the GV ubiquitin-related genes may function to regulate programmed cell death of the infected host cells. In the mesopelagic layer, which is characterized by relatively low host biomass (or encounter rates), the ability to suppress apoptosis might represent an adaptive strategy to prolong the life of the infected host cells, thereby allowing for greater viral production before host lysis and ensuring the continued propagation of the GV in resource-scarce environments.

## Putative hosts of meso-exclusive GVMAGs

To gain insights into the potential hosts of the 48 GVMAGs, we first performed homology searches against the MarFERReT database (Fig. S11). The best eukaryotic matches differed between meso-exclusive and other GVMAGs, with the former displaying fewer matches to green algae (i.e., Mamiellophyceae and Chlorophyceae). Meso-exclusive GVMAGs were enriched in matches to Prymnesiophyceae, Dinophyceae, and Mediophyceae. Of other GVMAGs, MV5 and NCV18 showed many matches to a single taxon (Fig. S11). MV5 showed four genes with best matches to Cryptophyceae, which was previously predicted as a host group of mirusvirus (6). NCV18, which is closely related to Emiliania huxleyi viruses, showed many matches to *Emiliania huxleyi* (class Prymnesiophyceae) probably due to the inclusion of viral sequences or homologs in the MarFERReT database.

We conducted phylogenetic analyses of protein sequences from meso-exclusive GVs that showed significant similarities with eukaryotic protein sequences (Fig. S12). We found three cases that supported the predictions by the above MarFERReT-based analysis. The NifU-like N terminal domain (NifU_N) of MV2 was clearly related to eukaryotic homologs with the closest relationship with sequences from dinoflagellates (*Kryptoperidinium*) and stramenopiles. Two other cases suggested relationships between meso-exclusive GVs and specific eukaryotes. However, interpreting these two trees as a support for specific virus-host relationships is not straightforward as most of other related sequences were bacterial sequences.

We further conducted host inference by examining co-occurrence patterns between meso-exclusive GVs and microeukaryotes. Using metatranscriptome and 18S rRNA data from mesopelagic samples, we calculated Spearman's correlation coefficients for the relative abundance between meso-exclusive GVs and ASVs in pico- or total-fraction. Nine ASVs exhibited statistically significant positive correlations with five meso-exclusive GVs ($P$-value < 0.05; Fig. S13). The predicted potential hosts were heterotrophs (or potentially mixotrophs), belonging to Cercozoa (3 ASVs; Cercozoa, Cercozoa_X, Filosa-Imbricatea; amoeboids or flagellates of Rhizaria), Picozoa (1 ASV; heterotrophic picoeukaryotes), Dinophyceae (1 ASV), and Syndiniales (4 ASVs; early branching dinoflagellates). These potential hosts were undetected in the 18S rRNA/rDNA data from the surface layer, suggesting mesopelagic or deeper layers as their main habitat. As members of Syndiniales parasitize dinoflagellates, their co-occurrence with GVs may reflect shared hosts rather than host-virus relationships. Cercozoa, Picozoa, and Dinophyceae were reported in the mesopelagic layer, with Cercozoa in particular showing the highest activity among

all depths (43). Host–virus associations have been suggested for Cercozoa and Dinofla-gellata based on single-cell genomics data from induced algal bloom samples (24).

Although these analyses point to potential hosts of meso-exclusive GVs, the limited representation of mesopelagic protist genomes in current reference databases, the small number of samples in this study, and the low abundance of GVs in the mesopelagic layer mean that these host predictions should be considered preliminary. Other approaches such as single-cell genomics may directly link GVs to their hosts, offering validation beyond sequence homology and co-occurrence (24).

## Deep-sea-specific GVs distribute widely in the global ocean

Our discovery of meso-exclusive GVs at a coastal site in Japan raises questions regarding their distributions and adaptation across different oceans and depth ranges. To address this issue and to catalog GV genomes specific to aphotic layers of the ocean, we constructed the GVGR database (see Materials and Methods) (Table S8), containing 4,473 GVMAGs and analyzed 1,890 publicly available marine metagenomes collected from depths ranging from 0 m to 10,899 m (Fig. S14).

The GVMAGs were found to have a wide vertical distribution, extending to a depth of 5,601 m (Fig. 3a). Based on the occurrence of GVMAGs across depths, we identified 101 deep-sea-specific GVMAGs that were only or predominantly distributed in the mesopelagic or deeper layers (Fig. 3b; Table S8) (see Materials and Methods). To further investigate their biogeographic patterns, we analyzed the distribution of deep-sea-specific GVMAGs across eight geographic regions and quantified their geographic ranges. The total number of deep-sea-specific GVMAGs was relatively high in the North Atlantic Ocean, North Pacific Ocean, and Arctic Oceans, ranging from 44 to 53 (Fig. 3c). Only eight were detected in the Southern Ocean, which may be due to limited sampling efforts.

The connectivity map revealed sharing of GVMAGs between multiple oceanic regions (Fig. 3c), suggesting the existence of widely distributed deep-sea-specific GVMAGs. Indeed, 55.4% of the deep-sea-specific GVMAGs were detected in two or more regions, with 10 GVs detected in five or more regions (Fig. 4d). The Arctic Ocean and Southern Ocean exhibited a large proportion of unique deep-sea-specific GVMAGs (36.4% and 37.5%, respectively). These polar oceans, thus, represent a "hotspot" of endemic deep-sea-specific GVMAGs, consistent with previous findings (19), and may be explained by limited water mass exchanges and steep environmental gradients that forms strong ecological barriers between high and lower latitude regions.

Phylogenetic analysis revealed that identified deep-sea-specific GVMAGs are scattered across the tree (Fig. 4a), reminiscent of the phylogenetic distribution of GVs specific to cold environments (20). One clade within the order *Imitervirales* was enriched with deep-sea-specific GVMAGs (Fig. 4b); two Muroto meso-exclusive GVMAGs (NCV01 and NCV26) were also placed within this clade. The members of this clade showed high relative abundances (Fig. 4c) and wide geographic distributions (Fig. 4d). The Muroto meso-exclusive NCV01 (represented by ERS493705_165 after dereplication) was distributed in four oceanic regions (Fig. 3c and 4d).

To investigate potential functional adaptations of deep-sea-specific GVMAGs, we identified significantly enriched KO terms and Pfam domains in deep-sea-specific GVMAGs using Fisher's exact test with FDR correction ($P$-value < 0.05) (see Materials and Methods; Table S9). Seventy-six KO terms were found to be enriched in deep-sea-specific GVMAGs (Fig. S15 and S16). Consistent with our results from the Muroto data, genes related to the ubiquitin system were found as deep-sea-specific KOs. Ammonium transporters and glutaminase were also deep-sea-specific. A previous study also revealed an enrichment of ammonium transporters in GVs detected below 200 m and suggested that these transporters may enhance nitrogen acquisition by their hosts, possibly in competition with Thaumarchaeota (30). The above mentioned *Imitervirales* deep-sea clade showed the highest number of deep-sea-specific KO terms (Fig. S16), suggesting lineage-specific specialization. Pfam domain analysis revealed 74 domains enriched in deep-sea-specific GVMAGs (Table S9), including homologs of cytochrome P450, CTP

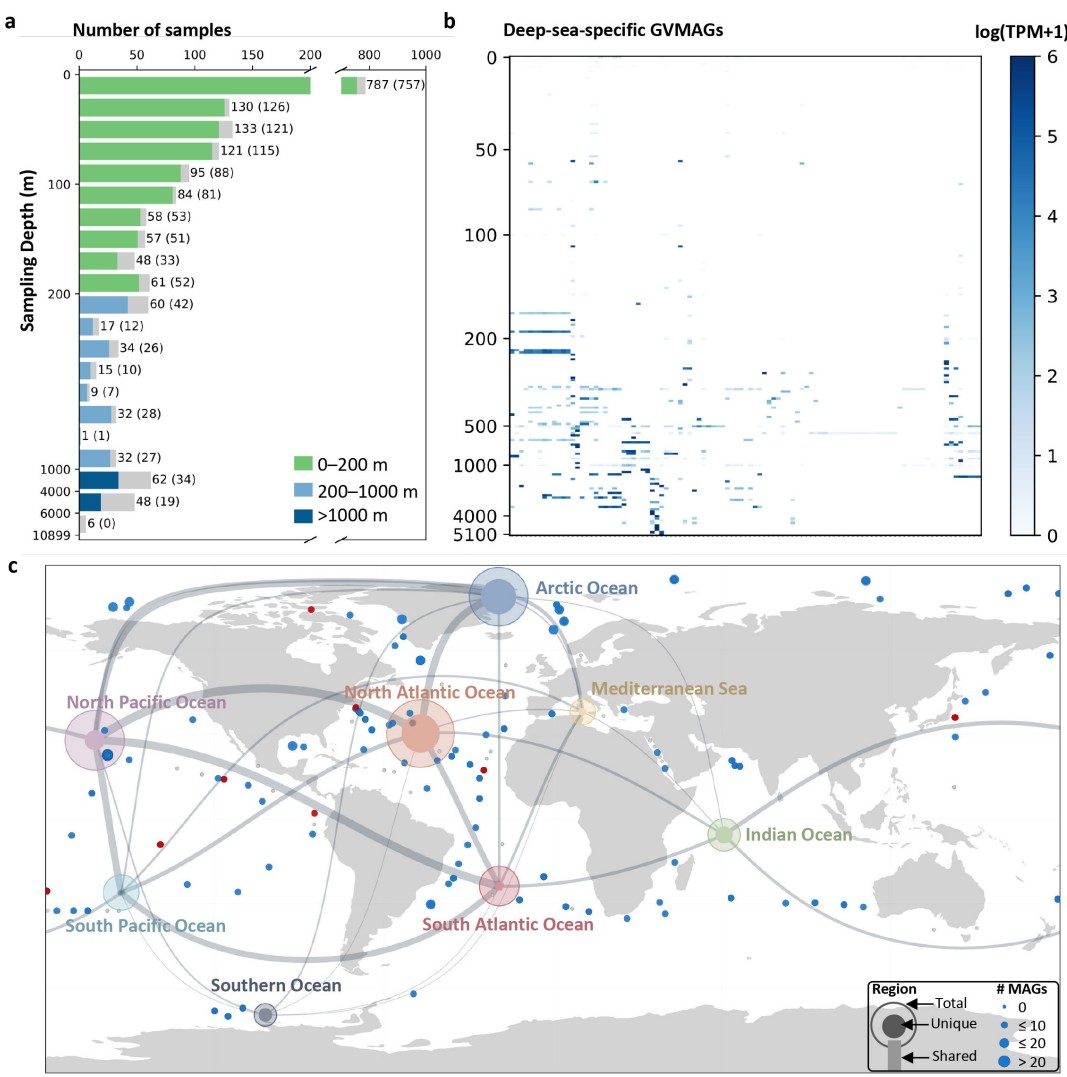

**FIG 3** Distribution of all GVMAGs in the GVGR database and 101 deep-sea-specific GVMAGs across global ocean depth layers. (a) Total number of samples and number of samples with GVMAGs detected across different depths. Bars are color-coded by depth category. Numbers at the end of each bar indicate the total number of samples in that depth range; values in parentheses represent the number of samples in which GVMAGs from the GVGR database were detected. (b) Relative abundance of 101 deep-sea-specific GVMAGs identified in the GVGR database across all depth layers. Each row represents a depth, and each column represents one deep-sea-specific GVMAG, with darker blue indicating higher abundance levels. (c) Distribution of deep-sea-specific GVMAGs across global deep-sea samples. Each dot on the map represents a deep-sea sample, with its size indicating the number of deep-sea-specific GVMAGs detected. Gray points indicate samples without deep-sea-specific GVMAGs; blue points indicate samples containing deep-sea-specific GVMAGs; red points indicate samples in which ERS493705_165 (representing Muroto meso-exclusive NCV01 after 95% sequence identity clustering) was detected. Eight large circles correspond to oceanic regions; their sizes reflect the total number of deep-sea-specific GVMAGs, while concentric rings indicate the proportions of GVMAGs unique to the region. Lines connecting the circles represent GVMAGs shared between regions, with line thickness proportional to the extent of sharing.

synthases, S-adenosylmethionine decarboxylase, and NifU_N. P450 has been previously reported in many GV genomes and has been proposed to be functionally connected to the 2-oxoglutarate and Fe (II)-dependent dioxygenase (54), which was also enriched in deep-sea-specific GVMAGs. The function of P450 in GVs remains unknown; however, it has been hypothesized that these genes might modulate viral or host lipid pools to aid energy production (55). Additionally, NifU is involved in Fe-S cluster formation, which plays roles in diverse cellular processes. NifU has been previously reported in eukaryotes, although its specific function remains unresolved (56). Two KO terms and 22 Pfam domains were enriched in the non-deep-sea-specific GVMAGs (Table S9). These included domains for DNA damage repair (UV damage endonuclease and 5′–3′ exonucleases),

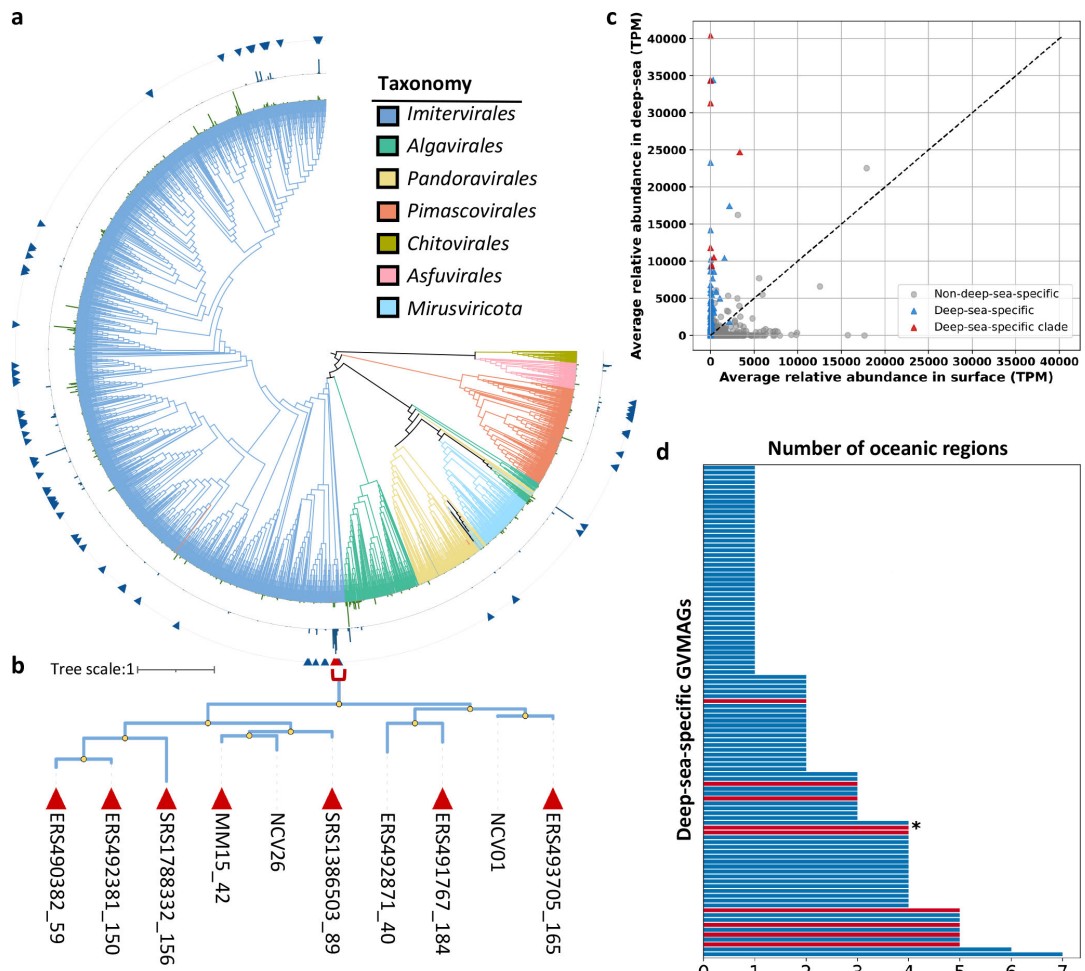

FIG 4  Phylogenetic diversity, abundance, and biogeography of deep-sea-specific GVMAGs. (a) Maximum-likelihood phylogenetic tree constructed from dereplicated GVGR genomes with 85% ANI, supplemented with 101 deep-sea-specific GVMAGs and Muroto meso-exclusive GVMAGs. Branch colors indicate taxonomic classifications. Green bars in the inner ring indicate the average relative abundances (TPM) of the GVMAGs in surface samples, and dark blue bars in the second ring indicate their average relative abundances in deep-sea samples. Triangles on the third ring indicate deep-sea-specific GVMAGs. The red brace highlights a clade composed predominantly of deep-sea-specific GVMAGs, as described in the main text. (b) Enlarged view of the deep-sea clade in the phylogenetic tree. Branches with ultrafast bootstrap support values > 95% are marked with yellow circles. The full identifier for "MM15_42" is Moniruzzaman_MM15_ERX556043_42_dc. NCV01 and NCV26 are Muroto meso-exclusive GVMAGs, with NCV01 being dereplicated with the deep-see-specific ERS493705_165 and NCV26 not recognized as deep-sea-specific. (c) Comparison of average relative abundances of GVMAGs in surface and deep-sea metage-nomes. Blue triangles denote deep-sea-specific GVMAGs, red triangles indicate those in the deep-sea clade (highlighted in [b]), and gray circles represent non-deep-sea-specific GVMAGs. (d) Number of oceanic regions in which each deep-sea-specific GVMAG was present. Red bars mark MAGs from the deep-sea clade, and the asterisk marks ERS493705_165 (representing Muroto NCV01).

rhodopsins, phosphate transport associated proteins (PhoH), and sulfotransferases. The enrichment of PhoH suggests adaptation to phosphate-limited conditions (57, 58).

Taken together, the KO and Pfam enrichment patterns revealed that GVs in the deep-sea have evolved to encode several genes of unique functions, potentially enabling them to maintain replicative efficiency by modulating host metabolic programs under environmental constraints such as limited light availability and scarce organic compounds.

## Summary

Our study provided the first integrative multi-omics evidence for the persistent activity of GVs in a mesopelagic environment. Six GVMAGs from the Muroto metagenomes were exclusively detected in the mesopelagic layer, many showing persistent transcriptional

activity across seasons. Some GVMAGs detected in the mesopelagic layer were active in the surface layer but inactive in the mesopelagic layer, suggesting that GV populations detected among mesopelagic metagenomes may partially originate from the surface through vertical transport. Furthermore, we identified 101 deep-sea-specific GVMAGs from a global metagenomic data set. Our results suggest that multiple lineages of GVs have independently adapted to deep-sea environments and are widely distributed across the global ocean. We also revealed clear genomic difference between deep-sea-specific and other GVs, including 76 KO terms and 74 Pfam domains enriched in the deep-sea-specific GVMAGs. Deep-sea-specific GVs exhibited distinct gene content, notably including genes for the ubiquitin system. These findings collectively support the hypothesis that distinct GV lineages may have evolved to adapt to deep-sea environments by acquiring genes to manipulate host cellular processes to thrive in the aphotic environment, which largely differs from that of the photic zone.

## MATERIALS AND METHODS

### Sample collection

Eight time-series seawater samples were collected from two depths (surface: 0.5 m, mesopelagic: 320 m) at the Kochi Prefectural Deep Seawater Laboratory, Kochi Prefecture, Japan (Fig. S1a), from December 2021 to April 2024. The seawater was pumped directly from the ocean and obtained via the deep-sea water intake system at the facility (37). The filtration system with membranes directly connected to the mesopelagic water tap was built to ensure that the water remained in its original condition (Fig. S1b; Table S1). After pre-filtration through a 150-µm-pore-size nylon mesh to remove large organisms, a large volume of seawater, especially for mesopelagic samples (30–1,110 L), was filtered through 3-µm- or 5-µm-pore-size polycarbonate membranes (Merck, Germany) and 0.2-µm-pore-size Sterivex-GP PES filters (Merck). Three groups of size fractionations were performed (Table S2). The first group (3.0–150 µm and 5.0–150 µm) mainly corresponds to nano- and microplankton and is referred to as the "nano/micro-fraction." The second group (0.2–150 µm) covers pico-, nano-, and microplankton and is referred to as the "total-fraction." The last group (0.2–3.0 µm and 0.2–5.0 µm) mainly corresponds to picoplankton and is referred to as the "pico-fraction." We collected pico-fraction samples and nano/micro-fraction samples in early samplings (until September 2022), and total-fraction samples and nano/micro-fraction samples in later samplings. All filtrated membranes were immediately frozen and transported in a cryo-shipper at liquid nitrogen temperatures to the laboratory and stored at −80°C until DNA and RNA were extracted. Filtration and cryo-preservation of each filter were completed within 25 min.

Total DNA and RNA were extracted using the AllPrep RNA/DNA kit (Qiagen, Germany) following the protocol described by Okazaki et al. (59). The DNA/RNA quantity was assessed using a Qubit fluorometer (Thermo Fisher Scientific, USA). The extracted DNA/RNA was used for 18S rDNA metabarcoding (28 samples), 18S rRNA metabarcoding (28 samples), metagenomic (13 samples), and metatranscriptomic (25 samples) sequencing (Fig. S1c). Environmental variables, including temperature, salinity, and pH, were recorded continuously by the Kochi Prefectural Deep Seawater Laboratory (Table S10).

### Metabarcoding sequencing and analyses

Microeukaryote amplicon sequencing was performed for both DNA and RNA-derived cDNA (rDNA and rRNA sets, respectively). For RNA, genomic DNA removal and first-strand cDNA synthesis were conducted using the SuperScript IV VILO Master Mix with the ezDNase enzyme kit (Thermo Fisher Scientific) following the manufacturer's protocol. The V4 region of the 18S rRNA gene from both cDNA and DNA was amplified by the KAPA HiFi HotStart ReadyMix (Roche) using the universal eukaryotic primer set E572F/

E1009R (60) attaching Illumina overhang adapters. The PCR conditions were as follows: initial pre-denaturation at 95°C for 2 min, followed by 30 cycles of denaturation at 98°C for 20 s, annealing at 61°C for 15 s, and extension at 72°C for 30 s, with a final extension at 72°C for 2 min. Triplicate PCR products were pooled and then purified using VAHTS DNA Clean Beads (Vazyme). The amplicon libraries were sequenced on the Illumina MiSeq platform to generate paired-end reads (2 × 300 bp).

Raw sequencing reads were processed using QIIME2 v2024.10 (61) with the DADA2 plugin (62). The "dada2 denoise-paired" command was used for adapter removal, primer removal, low-quality read trimming, dereplication, chimera removal, and amplicon sequencing variant (ASV) identification. Rare ASVs that appeared in less than two sequences across all samples were excluded. Taxonomic classification was performed on the remaining ASVs based on a pre-trained naive Bayes classifier trained against the PR2 reference database v5.0.0 using the "feature-classifier classify-sklearn" plugin (63). ASVs classified as non-protist lineages (i.e., metazoa, fungi) were removed. Finally, 1,616,977 reads from 56 samples were grouped into 5,031 ASVs. To enable comparisons between samples, the ASV table was rarefied to the minimum read depth per sample (7,581 reads), resulting in a final data set of 4,981 ASVs and 424,536 reads. Statistical differences among sample groups (e.g., depth and season) were tested by PERMANOVA (9,999 permutations) on the Bray–Curtis dissimilarity calculated from the Hellinger-transformed ASV abundance table, using the Vegan v2.6.10 package in R (64). The standardized Levins index (BA) (65, 66), representing the niche breadth, was calculated using the R package "spaa." For each depth and size fraction, BA was estimated from the relative abundance of ASVs in the surface and mesopelagic samples collected on the same day.

## Metagenomic sequencing and analyses

Metagenomic shotgun sequencing libraries were constructed using the Illumina DNA Prep kit (Illumina) and sequenced on the Illumina NovaSeq 6000 platform, generating 150 bp paired-end reads with an average of 58 Gbp per sample. Raw reads were quality-controlled using fastp v0.23.4 (67) with default parameters. For each metagenome, trimmed reads were assembled into contigs using MEGAHIT v1.2.9 (68) in "meta-sensitive" mode. Contigs over 2.5 kb were retained for downstream analysis. Trimmed reads from all samples were primarily used for cross-mapping against contigs assembled from individual samples by Bowtie2 v2.4.5 (69). Additionally, a co-assembly was performed by pooling reads from all mesopelagic samples. Contigs were clustered into bins using MetaBAT2 v2.15.15 (70). Metagenomic bins were generated for each of the 13 metagenomes and for the pooled mesopelagic metagenomes. Genes in MAGs were predicted using Prodigal-gv v2.11.0 (71, 72) with "meta" mode for marker gene identification and functional annotation.

NCV-MAGs and MV-MAGs were reconstructed separately following previously published methods (23, 33). To identify NCV-MAGs, we used a custom pipeline named hedera v.0.0.5 (https://github.com/banhbio/hedera). Briefly, the pipeline employs a gene density index (NCLDV index [23]) calculated using the genome size and the presence of 20 marker genes selected from the Nucleo-Cytoplasmic Virus Orthologous Groups (NCVOG) (73). MAGs with an NCLDV index over 5.75 are identified as potential NCV-MAGs. The NCV verification and decontamination is performed using the results of Viralrecall v2.1 (score > 0) (74), Virsorter2 v2.2.3 (max score group "NCLDV") (75), CAT v5.2.3 ("Nucleocytoviricota") (76), and hidden Markov models (HMMs) built with 149 NCVOGs (E-value $<1.0 \times 10^{-3}$) (23). Bins selected by hedera were further investigated for the possibility of chimeric bins (e.g., multiple viral genomes) by manual inspection using Anvi'o v8 (77). First, we primarily focused on deep branching clades that showed markedly different occurrence patterns or GC contents from other clades. If such clades showed the seven core marker genes of GVs (PolB, A32 packaging enzyme, superfamily II helicase, VLTF3 transcriptional factor, topoisomerase family II, transcription factor IIB, and RNA polymerase second largest subunit [RNAPS]), they were retained as a different bin; otherwise, they were discarded. Second, for the remaining clades, if we identified

clearly different patterns of occurrence or GC content, we split the bins into two or more sub-bins. Finally, the modified bins were re-assessed for the NCLDV index and only those with an NCLDV index over 5.75 were retained. For MV-MAGs, bins were identified as a mirusvirus if the HK97 MCP gene was detected using HMMER v3.4 (E-value <1.0 × 10$^{-3}$, bit score >100) (78). After manual inspection with Anvi'o, MAGs containing a contig encoding HK97-MCP were retained. Decontamination was performed using CheckV v1.0.1 (database v1.4) (79) by concatenating all contigs of each MAG and using "end_to_end" mode to remove prokaryotic contigs and provirus segments (33). Dereplication was conducted using dRep v3.5.0 (80) at an average nucleotide identify (ANI) of 95% with the parameters "--ignoreGenomeQuality --S_algorithm ANImf -sa 0.95 --clusterAg single -sizeW 1". The resulting non-redundant Muroto GVMAGs (NCV-MAGs and MV-MAGs) were species-level representatives. GVMAG names were assigned as follows: "NCV" or "MV," followed by a serial number indicating the rank of genome size from large to small (e.g., NCV10, MV5).

To determine the relative abundances of Muroto GVMAGs, the coverage and transcripts (or reads) per million (TPM) (81) of each MAG were calculated by mapping all quality-controlled trimmed reads to Muroto GVMAGs using CoverM v0.6.1 (82).

## Metatranscriptomic sequencing and analyses

RNA samples were subjected to metatranscriptomic sequencing (Fig. S1c). Poly-A strand-specific RNA sequencing libraries were constructed using the NEBNext Poly(A) mRNA Magnetic Isolation Module and NEBNext Ultra II Directional RNA Library Prep Kit (New England Biolabs, USA). Libraries were sequenced on the Illumina NovaSeq 6000 platform and 150 bp paired-end reads were generated. An average of 24 Gbp of sequence per sample was obtained. Raw reads were quality-controlled using fastp v0.23.4 with the "-l 100" parameter to remove low-quality reads and reads shorter than 100 bp. SortMeRNA v4.3.6 (83) was used to filter out rRNA reads against the Silva v138.1 LSU NR99, Silva v138.1 SSU NR99, RFAM 5s, and RFAM 5.8s databases using default settings. Filtered mRNA reads were mapped against the Muroto GVMAGs using Salmon v1.10.1 (84) with the parameters "--meta --minScoreFraction 0.95 --validateMappings". Transcript abundance was normalized as TPM. The results were used to quantify the relative expression level of each gene and MAG.

## Compiling the GVGR database and identifying deep-sea-specific GVMAGs

To investigate the global deep-sea GV distribution and adaptation, we compiled a comprehensive giant virus genome reference, named GVGR, by integrating the Muroto GVMAGs with publicly available GV metagenomic data sets (Fig. S4). We first extracted NCV-MAGs from metagenomic bins generated using MetaBAT2 by the OceanDNA MAG project (39). Metagenomic bins with an NCLDV index over 5.75 were considered as potential NCV-MAGs. These MAGs were decontaminated through retaining contigs identified as "NCLDV" by Virsorter2 v2.2.4 or those with a ViralRecall v2.1 score greater than 0. Host-derived sequences and proviral segments were removed using CheckV v1.0.1, as mentioned above. For MV-MAG classification, the initial screening was consistent with the process described for Muroto MV-MAGs. Contigs classified as "chromosome" or "plasmid" using geNomad v1.8.0 (72) were excluded. MAGs with a size between 50 kbp–3 Mbp in length were retained as GVMAGs.

Additionally, we incorporated publicly available GV genomes from multiple sources, including GOEVdb (2), 1,065 GVMAGs from Uranouchi Inlet, Japan (23), 293 LBGVMAGs from Lake Biwa (33), and 4 circular endogenous *Mirusviricota* (6, 40). All collected MAGs were pooled and dereplicated using dRep v3.5 at an ANI of 95% as previously mentioned. Within the ANI-based cluster, the MAGs with the highest N50 value among the top 30% largest sized genomes were selected as representative.

To assess MAG quality, a phylogeny-informed MAG assessment (PIMA) based on a phylogenetic tree and orthologous groups (OGs) was performed on the above-mentioned data set and Muroto GVMAGs (23). For phylogenetic tree reconstruction, seven

conserved marker genes were identified using the script "ncldv_markersearch.py" (57). For MV-MAGs, four marker genes (HK97-MCP, RNAPS, RNA polymerase largest subunit [RNAPL], and PolB) were identified using HMMER v3.4 (78) with pairwise alignment against the reference HMM profiles. These marker genes were aligned and concatenated using Clustal Omega v1.2.4 (85), trimmed with trimAl v1.2.1 (-gt 0.1) (86), and a phylogenetic tree was built with FastTree v2.1.11 (87) using default parameters, serving as an input for PIMA of NCV-MAGs and MV-MAGs, separately. OGs were classified using GVOG HMM profiles (88) using HMMER v3.4 ($E$-value < $1.0 \times 10^{-10}$) (78). If multiple OGs were assigned to a single sequence, the OG with the lowest $E$-value was retained.

PIMA was applied at a relative evolutionary divergence threshold of 0.65, corresponding to genus- or family-level classification (23). MAG consistency was evaluated based on the proportion of core genes present, while redundancy was calculated as the proportion of duplicated core genes exceeding the mode copy number observed across all MAGs within each lineage. MAGs exhibiting redundancy over 50% were removed. A second round of dRep dereplication was subsequently performed to integrate Muroto GVMAGs, producing the final GVGR data set. The taxonomy of GVMAGs in the GVGR data set was validated using TIGTOG (89). Gene prediction and abundance analyses in all 1,890 global samples (39) were performed using methods identical to those described above.

Each of these GVMAGs was categorized as "deep-sea-specific" or "non-deep-sea-specific" We considered genomes showing overrepresentation in deep-sea samples (depth > 200 m) as deep-sea-specific GVMAGs. The overrepresentation was ascertained using Mann–Whitney $U$ tests with Benjamini–Hochberg (BH) corrected $P$ values ($P$-value < 0.05) (90). We also considered genomes with signals only in deep-sea samples (TPM > 0) as deep-sea-specific GVMAGs. Other GVMAGs, not assigned to deep-sea-specific category, were referred to as "non-deep-sea-specific."

## Protein prediction and annotation

Proteins predicted from Muroto GVMAGs and the GVGR database were annotated with KEGG orthology (KO) using KofamScan v1.3.0 (91). KO annotations with the lowest $E$-values were retained. We further generated Pfam annotations of the genomes using AnnoMazing (https://github.com/BenMinch/AnnoMazing), a pipeline to annotate proteins based on HMM profiles. The annotations were performed using the Pfam database (92) with an E-value cut-off of $1.0 \times 10^{-5}$.

To identify KO terms and Pfam domains significantly enriched in deep-sea-specific GVMAGs, Fisher's exact test was conducted to compare the frequency of each annotated KO term in deep-sea-specific GVMAGs and non-deep-sea specific GVMAGs. KO terms with a total frequency ≤ 2 were excluded from further analysis.

To infer the potential eukaryotic hosts of GVs, we performed protein homology searches using BLASTP in DIAMOND v2.1.10 ($E$-value < $1.0 \times 10^{-5}$ and identity > 50%) (93) against the MarFERReT ver1.1.1 database (94). Hits with the lowest $E$-values were retained and used to assign candidate host taxonomy. For each retained protein, up to 100 homologs were further retrieved from a combined MarFERReT and NR database using DIAMOND BLASTP ($E$-value < $1.0 \times 10^{-5}$, identity > 50%). These homologous sequence sets were subsequently included in the phylogenetic analyses described below to assess putative host associations suggested by horizontal gene transfer.

## Analyses of phylogenetic diversity and clade delineation

A phylogenetic tree was first constructed based on PolB sequences (>500 amino acids) extracted from Muroto contigs (>1 kbp). The GOEV database and a wide range of eukaryotic and additional viral lineages (2, 95) were used as a reference set of PolB sequences to evaluate the completeness of GV recovery in Muroto GVMAGs. Another tree for *Mirusviricota* was generated using HK97-MCP sequences from Muroto MV-MAGs, combined with previously reported *Mirusviricota* genomes from the GOEV database, LBGVMAGs, and circular endogenous MV-MAGs. For *Nucleocytoviricota*, a concatenated

phylogenetic tree was constructed using the 7 core marker genes (88) for Muroto NCV-MAGs, along with 220 reference genomes from culture (2). Another concatenated phylogenetic tree was generated using the same 7 core marker genes, incorporating all deep-sea-specific MAGs, Muroto meso-exclusive MAGs, and the GVGR data set redundant by dRep v3.5.0 at an ANI of 85%. The marker gene search, alignment, concatenation, and trimming methods were the same as those described for the PIMA above. The tree was inferred using IQ-TREE v2.2.0 (96) with the ModelFinder Plus option to determine the best-fitting model. Support values were inferred using 1,000 ultrafast bootstraps (97). Tree visualization and rooting were carried out using iTOL v7 (98).

## ACKNOWLEDGMENTS

This study was supported by the Japan Society for the Promotion of Science (JSPS) KAKENHI (grant numbers 22H00384, 21H05057, 22H00385, 22H02420), and the Collaborative Research Program of the Institute for Chemical Research, Kyoto University (grant numbers 2025–32, 2024–34, 2022–32).

We thank the Kochi Prefectural Deep Seawater Laboratory for providing sampling permission and environmental metadata. We thank Hiroto Sasaki, Jun Xia, Qingwei Yang, Pascal Hingamap, and Keizo Nagasaki for their valuable assistance with sampling. Computational work was supported by the SuperComputer System, Institute for Chemical Research, Kyoto University. We thank Edanz (https://jp.edanz.com/ac) for editing a draft of this manuscript.

W.L. led the sampling, performed all experiments and most of the bioinformatics analyses. Y.N., J.W., K.N., and W.L. contributed to the compilation of the GVGR database. K.N. together with W.L. performed the statistical analysis on the GVMAGs from the GVGR database. H.B. contributed to the construction of the NCV-MAG pipeline. E.M. and L.M. contributed to part of the bioinformatics analyses. M.M. supervised E.M. T.Y. initiated the use of the Kochi Prefectural Deep Seawater Laboratory for this sampling. R.Y.N. contributed to the design of filtration systems. H.O. conceived the study and supervised W.L. and K.N. H.E. and Y.O. co-supervised W.L. and K.N. and participated in sampling. W.L. generated the initial draft of the manuscript. All authors contributed to the interpretation of data and writing of the manuscript, and all approved the final draft.

## AUTHOR AFFILIATIONS

[1]Bioinformatics Center, Institute for Chemical Research, Kyoto University, Uji, Japan

[2]Hatsukaichi Field Station, Fisheries Technology Institute, Japan Fisheries Research and Education Agency, Hatsukaichi, Japan

[3]Department of Marine Biology and Ecology, Rosenstiel School of Marine, Atmospheric, and Earth Science, University of Miami, Miami, Florida, USA

[4]Graduate School of Agriculture, Kyoto University, Kyoto, Japan

[5]Research Center for Bioscience and Nanoscience (CeBN), Research and Development Center for Marine Biosciences, Japan Agency for Marine-Earth Science and Technology (JAMSTEC), Yokosuka, Japan

## AUTHOR ORCIDs

Wenwen Liu  http://orcid.org/0009-0004-7260-2411
Lingjie Meng  https://orcid.org/0000-0001-9937-8860
Russell Y. Neches  http://orcid.org/0000-0002-2055-8381
Mohammad Moniruzzaman  https://orcid.org/0000-0001-9337-3874
Hiroyuki Ogata  http://orcid.org/0000-0001-6594-377X

## FUNDING

| Funder | Grant(s) | Author(s) |
|---|---|---|
| Japan Society for the Promotion of Science | 22H00384, 21H05057, 22H00385, 22H02420 | Hiroyuki Ogata |
| Institute for Chemical Research, Kyoto University | 2025-32, 2024-34, 2022-32 | Hiroyuki Ogata |

## AUTHOR CONTRIBUTIONS

Wenwen Liu, Data curation, Formal analysis, Investigation, Methodology, Validation, Visualization, Writing – original draft, Writing – review and editing | Komei Nagasaka, Data curation, Formal analysis, Investigation, Methodology | Junyi Wu, Data curation, Methodology | Hiroki Ban, Methodology, Software, Writing – review and editing | Ethan Mimick, Formal analysis, Validation | Lingjie Meng, Methodology, Writing – review and editing | Russell Y. Neches, Investigation, Methodology, Writing – review and editing | Mohammad Moniruzzaman, Formal analysis, Validation, Writing – review and editing | Takashi Yoshida, Methodology, Writing – review and editing | Yosuke Nishimura, Data curation, Resources, Writing – review and editing | Hisashi Endo, Investigation, Methodology, Supervision, Writing – review and editing | Yusuke Okazaki, Investigation, Methodology, Supervision, Writing – review and editing | Hiroyuki Ogata, Conceptualization, Funding acquisition, Investigation, Methodology, Project administration, Supervision, Writing – review and editing

## DATA AVAILABILITY

The data generated from the article, the processed data, and metadata are available in the DNA database of Japan (DDBI) at www.ddbj.nig.ac.jp and can be accessed under accession numbers PRJDB35423 and PRJDB35424. The nucleotide and protein sequences of the Muroto GVMAGs and GVGR database constructed in this study are available at GenomeNet: https://www.genome.jp/ftp/db/community/GVGR_MurotoGV. Other data used in this study include the Giant Virus Orthologous Groups (GVOGs) database (https://faylward.github.io/GVDB/).

## ADDITIONAL FILES

The following material is available online.

### Supplemental Material

**Supplemental Figures (mSystems00932-25-s0001.docx).** Fig. S1 to S16.
**Supplemental Tables (mSystems00932-25-s0002.xlsx).** Tables S1 to S10.

### Open Peer Review

**PEER REVIEW HISTORY (review-history.pdf).** An accounting of the reviewer comments and feedback.

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
