## [Reviewer comments · mSystems]

Giant viruses specific to deep oceans show persistent presence and activity

Wenwen Liu, Komei Nagasaka, Junyi Wu, Hiroki Ban, Ethan Mimick, Lingjie Meng, Russell Neches, Mohammad Moniruzzaman, Takashi Yoshida, Yosuke Nishimura, Hisashi Endo, Yusuke Okazaki, and Hiroyuki Ogata

Corresponding Author(s): Hiroyuki Ogata, Kyoto Daigaku

Review Timeline:

Submission Date:	June 25, 2025
Editorial Decision:	July 11, 2025
Revision Received:	October 2, 2025
Editorial Decision:	October 17, 2025
Revision Received:	October 20, 2025
Accepted:	October 22, 2025

Editor: Xiyang Dong

Reviewer(s): Disclosure of reviewer identity is with reference to reviewer comments included in decision letter(s). The following individuals involved in review of your submission have agreed to reveal their identity: Hanpeng Liao (Reviewer #1)

Transaction Report:

DOI: <https://doi.org/10.1128/msystems.00932-25>

Re: mSystems00932-25 (**Giant viruses specific to deep oceans show persistent presence and activity**)

Dear Prof. Hiroyuki Ogata:

Revision Guidelines

Sincerely,
Xiyang Dong
Editor
mSystems

Reviewer #1 (Comments for the Author):

This study systematically investigates the diversity, distribution, and transcriptional activity of GVs in the deep-sea environment using an integrative approach that combines 18S rDNA and rRNA metabarcoding, metagenomics, and metatranscriptomics. The authors identified 48 GVMAGs, six of which were exclusively detected in the mesopelagic layer and exhibited persistent transcriptional activity. In addition, the study constructed a GVGR to analyze the global distribution patterns of these viruses, revealing certain GV lineages that may be adapted to the deep-sea environment. This work provides a valuable regional dataset

for future studies of marine viral ecology. The methodological framework is comprehensive, incorporating multiple bioinformatic tools for data processing and validation, and the overall technical pipeline is sound. However, in the results section, the frequent shifts between depth layers may make it somewhat difficult for readers to follow the central findings. It is recommended that the structure of the results be further optimized to more clearly and coherently present the major observations from each depth, thereby highlighting the key messages of the study.

1. The current approach of identifying potential giant virus hosts based on ASVs detected in both 18S rDNA and rRNA exclusively in the mesopelagic layer is reasonable. However, additional evidence is still needed to support a confident host-virus association. While concurrent detection of rDNA and rRNA may suggest metabolic activity, it remains an indirect proxy for host identity unless validated by more robust host identification methods. Furthermore, if the objective is to demonstrate the stability or persistence of candidate hosts, it would be more appropriate to analyze the abundance patterns of those specific ASVs rather than presenting data from the entire eukaryotic community.

2. It may also be valuable to consider another group of ASVs that are present in both the surface and mesopelagic layers based on rDNA, but show transcriptional activity only in the mesopelagic zone based on rRNA. These ASVs may represent organisms that are ecologically restricted to the mesopelagic layer but appear in surface samples due to vertical mixing or advection. In such cases, their presence in surface rDNA does not necessarily reflect surface activity or ecological adaptation. Including these ASVs in the analysis, for example through comparisons of rRNA-to-rDNA ratios or the use of environmental modeling, could help identify additional candidate hosts that may otherwise be overlooked using strict presence-absence criteria.

3. Line 120-121, the ecological relevance of the "large" and "small" size fractions is unclear. Notably, the small size fraction includes 0.2-150 μm , which overlaps with the large size fraction (e.g., 3.0-150 μm or 5.0-150 μm), making the distinction ambiguous. Although this grouping appears in Figures 1 and 2, it is not addressed throughout the results section. If it is not central to the main findings, it may be worth simplifying or removing this part to improve clarity.

4. The overall research framework is conceptually sound, beginning with the identification of mesopelagic-specific GVMAGs from collected samples and subsequently extending the findings through the analysis of published datasets. However, the logic of the results section appears somewhat disjointed. While the seawater samples were collected from the mesopelagic layer, the GVGR database constructed from publicly available data is framed as representing the broader deep ocean. This creates a disconnect, particularly as the first part of the results focuses on GVMAGs from the authors' own samples but then shifts rather abruptly toward a more generalized deep-sea perspective in the latter part. A clearer linkage between these two components would improve the coherence of the narrative.

Reviewer #2 (Comments for the Author):

The manuscript provides a comprehensive multi-omics investigation into giant viruses (GVs) at both surface water and deep-ocean environments, addressing a knowledge gap in marine giant viral ecology. The integration of metabarcoding, metagenomic, and metatranscriptomic data from seasonal time-series sampling at Muroto (Japan), combined with a global meta-analysis, offers evidence for deep-sea-specific GV with activity and adaptive genomic features. The manuscript is generally well-structured, with clear figures and supplemental support. The abstract and introduction set the context well. Overall, the study advances understanding of GV ecology in the dark ocean. Some weaknesses include limited discussion of viral adaptive mechanisms, global distribution, and potential viral-host co-evolution, as well as uncertainty in host assignment.

Specific Comments

- Abstract: Specify "mesopelagic" instead of "deep-sea" for clarity, as sampling focused on only 320 m (mesopelagic).
- Functional Adaptations: The discussion about how enriched genes confer advantages in aphotic zones (e.g., energy conservation, stress response) is limited. For example, discuss how ubiquitin systems might counteract protein degradation in high-pressure/deep-sea conditions.
- Global distribution: 101 deep-sea-specific GVs spanning multiple lineages are identified; however, the discussion about biogeographic patterns of these deep-sea-specific GVs is lacking. Are deep-sea-specific GVs endemic or cosmopolitan?
- Host prediction: The homology-based host prediction is quite preliminary and unreliable. Phylogenetic analyses based on functional genes could give some hints. The authors may also consider discussing how future single-cell genomics (Hi-C) could aid in this process.
- In Fig. S5, PolB rings are confusing.

This study systematically investigates the diversity, distribution, and transcriptional activity of GVs in the deep-sea environment using an integrative approach that combines 18S rDNA and rRNA metabarcoding, metagenomics, and metatranscriptomics. The authors identified 48 GVMAGs, six of which were exclusively detected in the mesopelagic layer and exhibited persistent transcriptional activity. In addition, the study constructed a GVGR to analyze the global distribution patterns of these viruses, revealing certain GV lineages that may be adapted to the deep-sea environment. This work provides a valuable regional dataset for future studies of marine viral ecology. The methodological framework is comprehensive, incorporating multiple bioinformatic tools for data processing and validation, and the overall technical pipeline is sound. However, in the results section, the frequent shifts between depth layers may make it somewhat difficult for readers to follow the central findings. It is recommended that the structure of the results be further optimized to more clearly and coherently present the major observations from each depth, thereby highlighting the key messages of the study.

1. The current approach of identifying potential giant virus hosts based on ASVs detected in both 18S rDNA and rRNA exclusively in the mesopelagic layer is reasonable. However, additional evidence is still needed to support a confident host-virus association. While concurrent detection of rDNA and rRNA may suggest metabolic activity, it remains an indirect proxy for host identity unless validated by more robust host identification methods. Furthermore, if the objective is to demonstrate the stability or persistence of candidate hosts, it would be more appropriate to analyze the abundance patterns of those specific ASVs rather than presenting data from the entire eukaryotic community.
2. It may also be valuable to consider another group of ASVs that are present in both the surface and mesopelagic layers based on rDNA, but show transcriptional activity only in the mesopelagic zone based on rRNA. These ASVs may represent organisms that are ecologically restricted to the

mesopelagic layer but appear in surface samples due to vertical mixing or advection. In such cases, their presence in surface rDNA does not necessarily reflect surface activity or ecological adaptation. Including these ASVs in the analysis, for example through comparisons of rRNA-to-rDNA ratios or the use of environmental modeling, could help identify additional candidate hosts that may otherwise be overlooked using strict presence–absence criteria.

3. Line 120-121, the ecological relevance of the “large” and “small” size fractions is unclear. Notably, the small size fraction includes 0.2–150 μm , which overlaps with the large size fraction (e.g., 3.0–150 μm or 5.0–150 μm), making the distinction ambiguous. Although this grouping appears in Figures 1 and 2, it is not addressed throughout the results section. If it is not central to the main findings, it may be worth simplifying or removing this part to improve clarity.
4. The overall research framework is conceptually sound, beginning with the identification of mesopelagic-specific GVMAGs from collected samples and subsequently extending the findings through the analysis of published datasets. However, the logic of the results section appears somewhat disjointed. While the seawater samples were collected from the mesopelagic layer, the GVGR database constructed from publicly available data is framed as representing the broader deep ocean. This creates a disconnect, particularly as the first part of the results focuses on GVMAGs from the authors’ own samples but then shifts rather abruptly toward a more generalized deep-sea perspective in the latter part. A clearer linkage between these two components would improve the coherence of the narrative.

Green: Comments from reviewers

Blue: Responses

Reviewer #1 (Comments for the Author):

This study systematically investigates the diversity, distribution, and transcriptional activity of GVs in the deep-sea environment using an integrative approach that combines 18S rDNA and rRNA metabarcoding, metagenomics, and metatranscriptomics. The authors identified 48 GVMAGs, six of which were exclusively detected in the mesopelagic layer and exhibited persistent transcriptional activity. In addition, the study constructed a GVGR to analyze the global distribution patterns of these viruses, revealing certain GV lineages that may be adapted to the deep-sea environment. This work provides a valuable regional dataset for future studies of marine viral ecology. The methodological framework is comprehensive, incorporating multiple bioinformatic tools for data processing and validation, and the overall technical pipeline is sound. However, in the results section, the frequent shifts between depth layers may make it somewhat difficult for readers to follow the central findings. It is recommended that the structure of the results be further optimized to more clearly and coherently present the major observations from each depth, thereby highlighting the key messages of the study.

We thank the reviewer for his/her time and insightful comments on our manuscript. We greatly appreciate the positive feedback on the research scope and methodological framework, as well as his/her constructive suggestions for improving the clarity of the results section. In response, we have carefully revised the manuscript to minimize the confusion caused by frequent shifts in the descriptions of the results from different depth layers in the previous version of our manuscript.

Specifically, we simplified the presentation by removing detailed descriptions of the origin (surface or mesopelagic) of GVMAGs (LN162-168). Additionally, in responding to a comment from another reviewer, we consistently used “mesopelagic” when referring to the Muroto section, and “deep-sea” when discussing the global GV analyses to avoid confusing. We believe that these revisions have improved the coherence of the results section and clarify the key findings from the mesopelagic layer at Muroto and from the broader deep-sea samples at the global level.

Our detailed responses to specific comments are provided below.

1. The current approach of identifying potential giant virus hosts based on ASVs detected in both 18S rDNA and rRNA exclusively in the mesopelagic layer is reasonable. However, additional evidence is still needed to support a confident host-virus association. While concurrent detection of rDNA and rRNA may suggest metabolic activity, it remains an indirect proxy for host identity unless validated by more robust host identification methods. Furthermore, if the objective is to demonstrate the stability or persistence of candidate hosts, it would be more appropriate to analyze the abundance patterns of those specific ASVs rather than presenting data from the entire eukaryotic community.

Thank you for your valuable suggestion. Another reviewer also pointed out problems in our host inference. First, we removed the descriptions of the ASVs detected in both 18S rDNA and 18S rRNA, as the lack of detection by DNA (but positive detection by RNA) does not mean the absence of the organism corresponding to the ASVs.

Then, to objectively demonstrate the stability of the eukaryotic community in mesopelagic water in the Muroto sampling site, we added a comparison of Levins' standardized niche breadth between surface and mesopelagic samples collected on the same day (Fig. S3; method described in LN432-435). The significantly higher BA values were observed for mesopelagic ASVs compared to surface ASVs in both size fractions (Fig. S3, P -value < 0.001) indicate higher persistence for

mesopelagic ASVs. These results suggest that the mesopelagic environment may provide a unique and stable host community for GVs (LN142-150).

See below for the improved host inference analyses.

2. It may also be valuable to consider another group of ASVs that are present in both the surface and mesopelagic layers based on rDNA, but show transcriptional activity only in the mesopelagic zone based on rRNA. These ASVs may represent organisms that are ecologically restricted to the mesopelagic layer but appear in surface samples due to vertical mixing or advection. In such cases, their presence in surface rDNA does not necessarily reflect surface activity or ecological adaptation. Including these ASVs in the analysis, for example through comparisons of rRNA-to-rDNA ratios or the use of environmental modeling, could help identify additional candidate hosts that may otherwise be overlooked using strict presence-absence criteria.

Thank you again for the insightful comment. To improve the host inference, we added two new analyses, phylogenetic analyses (Fig. S12; LN266-274) and co-occurrence analyses (Fig. S13; LN275-290). Specifically, we tried to corroborate the host inference results obtained by the previous BLAST-based analysis (Fig. S11) using phylogenetic analyses. Our new results confirmed one case, while the results for two other cases were less clear (Fig. S12).

Co-occurrence analyses instead provided seemingly more promising host candidate ASVs that co-varies with GVMAGs. Nine ASVs exhibited statistically significant positive correlations with five meso-exclusive GVs (P -value < 0.05; Fig. S13). The predicted potential hosts were heterotrophs (or potentially mixotrophs), belonging to Cercozoa (3 ASVs; Cercozoa, Cercozoa_X, Filosa-Imbricatea; amoeboids or flagellates of Rhizaria), Picozoa (1 ASV; heterotrophic picoeukaryotes), Dinophyceae (1 ASV), and Syndiniales (4 ASVs; early branching dinoflagellates).

Regarding the vertical distribution of these candidate hosts as the review pointed out, we also examined their detections in the surface water samples (Fig. S13). These nine potential hosts were undetected in the 18S rRNA/rDNA data from the surface layer. This result suggests mesopelagic or deeper layers as their main habitat.

3. Line 120-121, the ecological relevance of the "large" and "small" size fractions is unclear. Notably, the small size fraction includes 0.2-150 μm , which overlaps with the large size fraction (e.g., 3.0-150 μm or 5.0-150 μm), making the distinction ambiguous. Although this grouping appears in Figures 1 and 2, it is not addressed throughout the results section. If it is not central to the main findings, it may be worth simplifying or removing this part to improve clarity.

Thank you for highlighting this. Indeed, the size fractions were not frequently mentioned in our work, but the comparison of the GV metatranscriptomic read mappings between the "large" and "small" fractions suggests that active infection of GVs in the mesopelagic layer is mainly associated with smaller host cells (Fig. S7, LN216-219). We consider this result is worth mentioning. Therefore, we decided not to omit the definition of size fractions in the results section.

We still agree that the terms "small" and "large" might be confusing. To improve clarity, we have now categorized the size fractions into three groups. The first group (3.0–150 μm and 5.0–150 μm) corresponds to nano- and microplankton and is referred to as the "nano/micro-fraction" in this study. The second group (0.2–150 μm) covers pico-, nano-, and microplankton and is referred to as the "total-fraction". The last group (0.2–3.0 μm and 0.2–5.0 μm) mainly corresponds to picoplankton and is referred to as the "pico-fraction" (LN116-118, LN388-395). Fig.1a-d, Fig.2bc and Fig.S1 were also updated. In addition, Fig. S7 now presents only the total-fraction and nano/micro-fraction to make the results easier to interpret.

4. The overall research framework is conceptually sound, beginning with the identification of mesopelagic-specific GVMAGs from collected samples and subsequently extending the findings through the analysis of published datasets. However, the logic of the results section appears

somewhat disjointed. While the seawater samples were collected from the mesopelagic layer, the GVGR database constructed from publicly available data is framed as representing the broader deep ocean. This creates a disconnect, particularly as the first part of the results focuses on GVMAGs from the authors' own samples but then shifts rather abruptly toward a more generalized deep-sea perspective in the latter part. A clearer linkage between these two components would improve the coherence of the narrative.

Thank you for the comment. To improve the transition, modified the first sentence of the result section “Deep-sea-specific GVs distribute widely in the global ocean” as follows (LN298-301):

“Our discovery of meso-exclusive GVs at a coastal site in Japan raises questions regarding their distributions and adaptation across different oceans and depth ranges. To address this issue and to catalog GV genomes specific to aphotic layers of the ocean, we constructed the GVGR database...”

Related to this, we modified a part of the text in the abstract (LN34-37): *“To further investigate the distribution and phylogenomic features of GVs at a global scale across broader depths, we compiled GV reference genomic data from the OceanDNA MAG project and other resources, and analyzed 1,890 marine metagenomes.”*

With these changes, we believe that the disconnect pointed out by this reviewer is now mitigated in the revised version of our manuscript.

Reviewer #2 (Comments for the Author):

The manuscript provides a comprehensive multi-omics investigation into giant viruses (GVs) at both surface water and deep-ocean environments, addressing a knowledge gap in marine giant viral ecology. The integration of metabarcoding, metagenomic, and metatranscriptomic data from seasonal time-series sampling at Muroto (Japan), combined with a global meta-analysis, offers evidence for deep-sea-specific GVs with activity and adaptive genomic features. The manuscript is generally well-structured, with clear figures and supplemental support. The abstract and introduction set the context well. Overall, the study advances understanding of GV ecology in the dark ocean. Some weaknesses include limited discussion of viral adaptive mechanisms, global distribution, and potential viral-host co-evolution, as well as uncertainty in host assignment.

We appreciate your constructive comments. We are grateful to have the opportunity to improve our manuscript based on your suggestions. We have expanded our analysis and discussion of potential viral-host associations, further elaborated the manuscript regarding deep-sea-specific functions, and added additional analyses regarding the global distribution of the deep-sea-specific GVs. We provide our responses to specific comments below.

Specific Comments

- Abstract: Specify "mesopelagic" instead of "deep-sea" for clarity, as sampling focused on only 320 m (mesopelagic).

Thank you for pointing this out. In the revised manuscript, we have replaced the term “deep-sea” with “mesopelagic” in the abstract. Additionally, throughout the Muroto sections, we now consistently use “mesopelagic” instead of “deep-sea”.

- Functional Adaptations: The discussion about how enriched genes confer advantages in aphotic zones (e.g., energy conservation, stress response) is limited. For example, discuss how ubiquitin systems might counteract protein degradation in high-pressure/deep-sea conditions.

We have introduced additional information on how ubiquitin systems may benefit GVs in adapting to aphotic conditions (LN241-253), which reads as follows:

“The sequential action of ubiquitin systems conjugates ubiquitin to proteins and target them to

proteasomes for degradation (47). Eukaryotic viruses can hijack the ubiquitin-proteasome system to aid in various stages of viral propagation (48). Since normal protein folding is hindered by low temperature or high pressure (49, 50), the ubiquitin-related genes of GVs may function to modulate the host ubiquitin-proteasome system to quality control viral and host proteins in harsh condition in the mesopelagic layer, as previously hypothesized for GVs in cold environments (51). As the ubiquitin-proteasome system is also known to regulate apoptosis (52), the GV ubiquitin-related genes may function to regulate programmed cell death of the infected host cells. In the mesopelagic layer, which is characterized by relatively low host biomass (or encounter rates), the ability to suppress apoptosis might represent an adaptive strategy to prolong the life of the infected host cells, thereby allowing for greater viral production before host lysis and ensuring the continued propagation of the GV in resource-scarce environments.”

In addition, we expanded the discussion of ammonium transporters (LN336-339):

“A previous study also revealed an enrichment of ammonium transporters in GVs detected below 200 m and suggested that these transporters may enhance nitrogen acquisition by their hosts, possibly in competition with Thaumarchaeota (29).”

- Global distribution: 101 deep-sea-specific GVs spanning multiple lineages are identified; however, the discussion about biogeographic patterns of these deep-sea-specific GVs is lacking. Are deep-sea-specific GVs endemic or cosmopolitan?

Thank you for pointing out this. To address, we have added analyses of the connectivity patterns of deep-sea-specific GVMAGs across multiple oceanic regions in Fig. 3c and quantified the number of oceanic regions in which each deep-sea-specific GVMAG was detected (Fig. 4d). We have also expanded the corresponding discussion of the biogeographic patterns of these deep-sea-specific GVs as below

(LN307-321): *“To further investigate their biogeographic patterns, we analyzed the distribution of deep-sea-specific GVMAGs across eight geographic regions and quantified their geographic ranges. The total number of deep-sea-specific GVMAGs was relatively high in the North Atlantic Ocean, North Pacific Ocean and Arctic Oceans, ranging from 44 to 53 (Fig. 3c). Only eight were detected in the Southern Ocean, which may be due to limited sampling efforts. The connectivity map revealed sharing of GVMAGs between multiple oceanic regions (Fig. 3c), suggesting the existence of widely distributed deep-sea-specific GVMAGs. Indeed, 55.4% of the deep-sea-specific GVMAGs were detected in two or more regions, with 10 GVs detected in five or more regions (Fig. 4d). The Arctic Ocean and Southern Ocean exhibited a large proportion of unique deep-sea-specific GVMAGs (36.4% and 37.5%, respectively). These polar oceans thus represent a “hotspot” of endemic deep-sea-specific GVMAGs, consistent with previous findings (19), and may be explained by limited water mass exchanges and steep environmental gradients that forms strong ecological barriers between high and lower latitude regions.”*

(LN326-329): *“The members of this clade showed high relative abundances (Fig. 4c) and wide geographic distributions (Fig. 4d). The Muroto meso-exclusive NCV01 (represented by ERS493705_165 after dereplication) was distributed in four oceanic regions (Fig. 3c, Fig. 4d).”*

- Host prediction: The homology-based host prediction is quite preliminary and unreliable. Phylogenetic analyses based on functional genes could give some hints. The authors may also consider discussing how future single-cell genomics (Hi-C) could aid in this process.

Thank you very much for these valuable suggestions. We agree that homology-based host prediction was preliminary. To improve the host inference, we added two new analyses, phylogenetic analyses (Fig. S12; LN266-274) and co-occurrence analyses (Fig. S13; LN275-290). Specifically, we tried to corroborate the host inference results obtained by the previous BLAST-based analysis (Fig. S11) using phylogenetic analyses. Our new results confirmed one case, while the results for two other cases were less clear (Fig. S12).

Co-occurrence analyses instead provided seemingly more promising host candidate ASVs that co-varies with GVMAGs. Nine ASVs exhibited statistically significant positive correlations with five meso-exclusive GVs (P-value < 0.05; Fig. S13). The predicted potential hosts were heterotrophs (or potentially mixotrophs), belonging to Cercozoa (3 ASVs; Cercozoa, Cercozoa_X, Filosa-Imbricatea; amoeboids or flagellates of Rhizaria), Picozoa (1 ASV; heterotrophic picoeukaryotes), Dinophyceae (1 ASV), and Syndiniales (4 ASVs; early branching dinoflagellates).

In addition, we have expanded the discussion to highlight the potential of single-cell genomics which may directly link GVs to their hosts (LN294-296): “*Other approaches such as single-cell genomics may directly link GVs to their hosts, offering validation beyond sequence homology and co-occurrence (24).*”

- In Fig. S5, PolB rings are confusing.

We simplified the figure by removing one of the three rings. Now, the inner ring shows PolB sequences likely to be of GV origin and found in the Muroto contigs (>2.5 kbp), and the outer ring represents PolB sequences found in qualified Muroto GVMAGs.

Re: mSystems00932-25R1 (**Giant viruses specific to deep oceans show persistent presence and activity**)

Dear Prof. Hiroyuki Ogata:

Revision Guidelines

Sincerely,
Xiyang Dong
Editor
mSystems

Reviewer #1 (Comments for the Author):

The revised manuscript reflects considerable improvement in clarity and structure. I have no further comments or recommendations at this time.

Reviewer #2 (Comments for the Author):

The authors have addressed most of my concerns; I only have a few minor suggestions.

L73. Citation needed

L166. Italic for *Emiliana huxleyi*

Green: Comments from reviewers

Blue: Responses

Reviewer #1 (Comments for the Author):

The revised manuscript reflects considerable improvement in clarity and structure. I have no further comments or recommendations at this time.

We sincerely thank the reviewer for his/her time and the positive feedback on our work.

Reviewer #2 (Comments for the Author):

The authors have addressed most of my concerns; I only have a few minor suggestions.

We appreciate your constructive comments and are grateful for the opportunity to further improve our manuscript based on your suggestions.

- L73. Citation needed

Thank you for pointing this out. In the revised manuscript, we have cited the paper *Danovaro et al., 2017 (LN72)*, which states that “*The deep ocean encompasses 95% of the oceans’ volume*”.

- L166. Italic for *Emiliana huxleyi*

Thank you for the comment. The term “*Emiliana huxleyi virus*” is a virus name. According to the rules of ICTV (International Committee on Taxonomy of Viruses; <https://ictv.global/faq/names>), a virus names should not be italicized, even when they include the name of a host species or genus. Therefore, we have kept “*Emiliana huxleyi viruses*” in plain text. We know there are different styles in writing virus names that can vary across authors or journals, but we prefer to follow the rules of ICTV.

Re: mSystems00932-25R2 (**Giant viruses specific to deep oceans show persistent presence and activity**)

Dear Prof. Hiroyuki Ogata:

Your manuscript has been accepted, and I am forwarding it to the ASM production staff for publication. Your paper will first be checked to make sure all elements meet the technical requirements. ASM staff will contact you if anything needs to be revised before copyediting and production can begin. Otherwise, you will be notified when your proofs are ready to be viewed.

Sincerely,
Xiyang Dong
Editor
mSystems